

# Does suicidal desire moderate the association between frontal delta power and psychological pain?

Esther L. Meerwijk and Sandra J. Weiss

Department of Community Health Systems, University of California, San Francisco, San Francisco, CA, United States

## ABSTRACT

Psychological pain frequently underlies thoughts of suicide. We investigated if recent suicidal desire moderated the association between potential neurophysiological markers and psychological pain assessed on the Psychache Scale (PS) and the Orbach & Mikulincer Mental Pain Questionnaire (OMMP). The OMMP specifically assesses current psychological pain that may more readily capture emotions present during recent suicidal desire. In contrast, the PS leaves the timeframe undefined. A secondary analysis was conducted of resting-state EEG data and heart rate obtained in adults with a history of depression. In simultaneous multiple regression models, while controlling for depressive symptoms, recent suicidal desire moderated associations with right-frontal EEG delta power ($\Delta R^2 = .07$, $p < .01$) and low-frequency heart rate variability (nonsignificantly) for pain assessed on the PS. No indication for moderation was found for pain on the OMMP. The relationship between the two measures of psychological pain was stronger for individuals with recent suicidal desire ($r = .75$, $p < .01$ vs. $r = .50$, $p < .05$). The findings suggest that, unless a respondent's psychological pain is recent and substantial, the PS may not capture the intensity of current psychological pain as effectively as the OMMP.

## INTRODUCTION

Psychological pain is a subjective experience that involves negative appraisal of self, and as such requires self-report instruments to be assessed (*Meerwijk & Weiss, 2011*; *Tossani, 2013*). Among the first self-report instruments of psychological pain were Shneidman's Psychological Pain Assessment Scale (*Shneidman, 1999*) and the Psychache Scale ([PS], *Holden et al., 2001*). These were followed by the Orbach & Mikulincer Mental Pain Questionnaire ([OMMP], *Orbach et al., 2003a*) and more recently the Mee-Bunney Psychological Pain Assessment Scale (*Mee et al., 2011*) and the Three-Dimensional Psychological Pain Scale (*Li et al., 2014*). Of these instruments, the PS and OMMP are probably the most frequently used to assess psychological pain.

While psychological pain is not unique to people with depression, ample research has found that people with depression report high levels of psychological pain and that

Corresponding author
Esther L. Meerwijk,
esther.meerwijk@gmail.com

hopelessness covaries with psychological pain (*Van Heeringen et al., 2010*; *Mee et al., 2011*; *Li et al., 2014*; *Xie et al., 2014*; *Cáceda et al., 2014*; *Troister, D'Agata & Holden, 2015*). We recently reported that psychological pain as assessed on the OMMP, but not the PS, in adults with a history of depression correlated with objective measures of resting-state neurophysiological parameters, in particular low-frequency heart rate variability (HRV) and right frontal EEG delta power (*Meerwijk, Chesla & Weiss, 2014*; *Meerwijk, Ford & Weiss, 2015*). As discussed in our previous publications, the relationship between psychological pain and these neurophysiological parameters may result from resting-state dysfunction of the sympathetic nervous system and increased sympathetic activity under conditions of greater psychological pain. Different findings when using these two psychological pain scales were unexpected because they purport to measure the same construct of psychological pain. In light of the association of neurophysiological markers with suicide ideation (*Iosifescu et al., 2008*; *Hunter et al., 2010*; *Song et al., 2011*; *Chang et al., 2012*), and the associations found between suicide ideation and psychological pain in depression (*Olié et al., 2010*; *Van Heeringen et al., 2010*), we questioned whether recent suicidal desire of respondents could moderate and thus explain the differential findings obtained with the PS and OMMP in adults with a history of depression. The OMMP specifically assesses current psychological pain that may more readily capture emotions present during recent suicidal desire. In contrast, the PS leaves the timeframe undefined, allowing respondents to consider their lifetime experience when assessing psychological pain. However, individuals with recent suicidal desire would likely assess their pain on the PS in light of their current feelings. For these individuals, we would expect the relationship between the OMMP and PS scores to be stronger. In addition, we would expect their PS scores to be more strongly associated with resting-state neurophysiological parameters. We present here a secondary analysis that tested the following hypotheses: (1) the correlation between OMMP and PS scores will be stronger among individuals who report a recent desire for suicide than among individuals who do not report recent suicidal desire, and (2) the correlation of the PS score with low-frequency heart rate variability and right frontal delta power in the resting state will be stronger among individuals who report a recent suicidal desire than among those who do not report recent suicidal desire.

Ample empirical evidence supports the notion that psychological pain is associated with suicidal thoughts and behaviors. Several studies observed a relationship between a history of suicide attempts and psychological pain, and found that psychological pain significantly predicted suicide attempter status (one or more vs. zero attempts), after controlling for depression and hopelessness (*Pereira et al., 2010*; *Troister & Holden, 2010*; *Patterson & Holden, 2012*). Significantly higher psychological pain was found in people who had a history of suicide attempts versus people who never attempted suicide (*Orbach et al., 2003b*; *Flamenbaum & Holden, 2007*; *Levi et al., 2008*), but no significant differences were also reported (*Pompili et al., 2008*). *DeLisle & Holden (2009)* found that psychological pain predicted reasons for attempting suicide, after controlling for depression and hopelessness, and *Olié et al. (2010)* found that patients who were hospitalized for a depressive episode and who had attempted suicide, reported high

current psychological pain more often than inpatients who had not attempted suicide. While recent adverse events may trigger psychological pain and precipitate a suicide attempt, research has also shown that adverse events during early life can predispose an individual to suicide attempts during adulthood (*Pompili et al., 2011*). *Levi et al. (2008)* did not find a significant effect of psychological pain in inpatients with regard to the seriousness of suicide attempts.

Other studies reported on the relationship between the broader concept of suicide ideation and psychological pain. Medium to strong positive correlations were found between suicide ideation and psychological pain in undergraduate students (*Lester, 2000*; *Holden et al., 2001*; *Leenaars & Lester, 2004*) and in patients admitted for a depressive episode (*van Heeringen et al., 2010*). In the only prospective studies of psychological pain to date (*Troister & Holden, 2012*; *Troister et al., 2013*), change in psychological pain was the only predictor of suicide ideation over two years for undergraduate students who were thought to be at high risk of suicide at baseline. In a different study of undergraduates, *Troister & Holden (2013)* also reported that psychological pain was the stronger predictor of suicide ideation, partially mediating the contributions of depression and hopelessness.

## METHODS

Details about the study design, study sample, and data collection were published elsewhere (*Meerwijk, Chesla & Weiss, 2014*; *Meerwijk, Ford & Weiss, 2015*). In short, we collected resting-state heart rate and EEG data in adults ($N = 35$) with a history of depression, who did not necessarily experience a depressive episode at the time of data collection. We intentionally enrolled participants who self-reported a current or past diagnosis of mood disorder, as this was expected to result in a broad range of psychological pain in the sample. Participants responded to flyers distributed to waiting rooms of a psychiatric hospital's outpatient department and psychological services centers in the San Francisco Bay Area of California, and to online advertisement on Craigslist from January to September 2012. A member of the research team scheduled a time for study participation for respondents who met eligibility criteria and who provided written informed consent. Neurophysiological data were recorded during 5-minute sessions while participants were sitting upright and had been instructed to keep their eyes closed, sit as still as possible, not focus on anything in particular and let their mind run free. The EEG set up (Biosemi ActiveTwo, sample rate 1,024 Hz) involved 34 electrodes placed according to the 10/20 system. Horizontal and vertical electro-oculograms were recorded from the outer canthi of the eyes and above and below the left eye. Two flat-type electrodes, one on the right collarbone and one on the lower-left abdomen, were used to measure heart rate. Table 1 shows sociodemographic characteristics of the sample. The Institutional Review Board at the University of California, San Francisco, approved the study (IRB# 11-07119).

### Psychological measures

Psychological pain was assessed on the PS and OMMP and the order of administration was alternated to prevent testing effects. The PS has 13 items with higher scores reflecting greater psychological pain. Nine items (factor 1) are scored on a frequency scale ranging

**Table 1  Sociodemographic characteristics by recent suicidal desire.**

| | Recent suicidal desire | | | |
| --- | --- | --- | --- | --- |
| | No ($n = 24$) | Yes ($n = 11$) | Total ($N = 35$) | |
| Mean age in years (SD) | 35.42 (11.41) | 34.18 (13.27) | 34.91 (11.60) | $t = 0.28$ |
| Gender | | | | $\chi^2 = 0.18$[a] |
| Men | 5 | 3 | 8 | |
| Women | 19 | 8 | 17 | |
| Ethnicity | | | | $\chi^2 = 1.32$ |
| White | 13 | 4 | 17 | |
| African American | 2 | 1 | 3 | |
| Hispanic | 2 | 2 | 4 | |
| Asian | 3 | 2 | 5 | |
| Other or mixed[b] | 4 | 2 | 6 | |
| Marital status | | | | $\chi^2 = 1.25$ |
| Single | 21 | 10 | 31 | |
| Married/in a relationship | 3 | 1 | 4 | |
| Highest education | | | | $\chi^2 = 0.89$ |
| High school | 3 | 2 | 5 | |
| Some college | 6 | 4 | 10 | |
| College graduate | 9 | 3 | 12 | |
| Graduate/professional school | 6 | 2 | 8 | |
| Employment status | | | | $\chi^2 = 1.66$ |
| Unemployed | 11 | 3 | 14 | |
| Occasionally employed | 6 | 5 | 11 | |
| Regularly employed | 7 | 3 | 10 | |

**Notes.**

[a] All $\chi^2$ tests included cells with expected counts less than 5.

[b] Other included native Hawaiian, Pacific Islander, Native American, and Alaska native.

from never to always and four items reflecting pain intensity (factor 2) are scored on a symmetrical scale ranging from strongly disagree to strongly agree. The instrument is well validated in diverse populations, including outpatients with depression, male prison inmates, students at risk for suicide, and homeless men (*Mills, Green & Reddon, 2005*; *Patterson & Holden, 2012*; *Troister & Holden, 2012*; *Li et al., 2014*; *Xie et al., 2014*). We found excellent internal consistency reliability with Cronbach's $\alpha = .92$. The OMMP assesses current psychological pain at the time of completing the questionnaire and worst-ever psychological pain, with higher total scores reflecting greater psychological pain. A 5-item response scale ranging between strongly disagree and strongly agree is used for all 44 items. The OMMP was shown to possess a high degree of validity in patients admitted for a suicide attempt, in university students and in the general population (*Orbach et al., 2003a*; *Orbach et al., 2003b*; *Levi et al., 2008*; *Reisch et al., 2010*; *van Heeringen et al., 2010*; *Soumani et al., 2011*; *Nahaliel et al., 2014*). We found excellent internal consistency reliability with Cronbach's $\alpha = .95$. Because the PS does not provide a comparison for worst-ever psychological pain, here we report on the current psychological pain dimension of the OMMP only.

The Beck Scale for Suicide ideation ([BSS], *Beck, Brown & Steer, 1997*) was used to assess suicide ideation during the week that preceded the study. Suicidal desire was derived from BSS item 4, which indicates whether participants had an active suicidal

desire. This item was identified as having high discriminating power for assessing elevated suicide risk (*De Beurs et al., 2014*).

Prior research has shown that psychological pain covaries with depression and hopelessness. We included measures of depression and hopelessness to describe the sample and to allow removal of the contribution of these constructs to the relationship with psychological pain during statistical analysis. Participants completed the Beck Depression Inventory (BDI) II (*Beck, Steer & Brown, 1996*) and the Beck Hopelessness Scale ([BHS], *Beck & Steer, 1993*).

### Data processing and analysis

Heart rate data (lower left abdomen minus collarbone electrode, to maximize the signal and reduce signal noise) were visually inspected for artifacts and ectopic heart beats. Beat-to-beat intervals were determined using standard functions available from the BioSig 2.61 package for GNU Octave. We used fast Fourier transformation to determine low-frequency HRV and high-frequency HRV. Automated artifact detection was used to process frontal EEG data (F3/F4). Artifact-free data were divided into nonoverlapping 2 s epochs. Subsequently, EEG power was determined in standard EEG frequency bands (delta: 0.5–4 Hz, theta: 4–8 Hz, alpha: 8–13 Hz, beta: 13–30 Hz, gamma: 30–100 Hz) and averaged per band across epochs. HRV and EEG power results were natural log transformed to obtain more normally distributed variables. Based on prior analysis regarding associations with psychological pain assessed on the OMMP and PS, we concentrated our analysis for hypothesis two on low-frequency HRV and right frontal EEG delta power (*Meerwijk, Chesla & Weiss, 2014*; *Meerwijk, Ford & Weiss, 2015*).

To test the moderating effect of recent suicidal desire, we created two groups, one group that indicated no suicidal desire and another group that indicated at least a weak suicidal desire (BSS item 4 score > 0). Zero-order correlations were determined and simultaneous multiple regressions conducted to test the moderating effect of suicidal desire on the correlation between OMMP and PS (hypothesis one) and on the relationship between neurophysiological parameters and the PS score (hypothesis two). As hopelessness did not contribute significant variance to the regression models, we report models that control for level of depression only. R version 3.1.2 was used for all analyses, and statistical significance was assumed at $p < .05$.

## RESULTS

When comparing participants who reported a recent desire for suicide ($n = 11$) and participants with no desire for suicide or no recent desire for suicide ($n = 24$), the two groups did not differ with respect to sociodemographic characteristics (see Table 1). Table 2 shows clinical symptoms by suicidal desire. Based on their BDI scores, the majority of participants experienced moderate to severe depression. We found significant differences in depression and hopelessness, with more severe symptoms in participants with recent suicidal desire. They were also more likely to report lifetime suicide attempts.

A strong positive correlation between PS and OMMP was observed in both groups (see Table 3), although considerably stronger for the group with recent suicidal desire

**Table 2  Clinical characteristics (mean or frequency) by recent suicidal desire.**

| | Recent suicidal desire | | | |
| | No ($n = 24$) | Yes ($n = 11$) | Total ($N = 35$) | |
|---|---|---|---|---|
| PS (SD) | 37.4 (9.5) | 47.6 (8.9) | 40.6 (10.4) | $t = -2.98^{**}$ |
| OMMP (SD) | 117.2 (26.8) | 132.6 (28.0) | 122.0 (27.7) | $t = -1.57$ |
| BDI (SD) | 24.4 (10.3) | 33.4 (9.8) | 27.2 (10.8) | $t = -2.43^{*}$ |
| BHS (SD) | 10.2 (5.0) | 15.1 (4.3) | 11.7 (5.3) | $t = -2.81^{**}$ |
| Years since $D_x$ (SD) | 7.2 (6.6) | 5.9 (5.4) | 6.9 (6.3) | $t = -0.52$ |
| Diagnosis | | | | $\chi^2 = 6.40^{*}$ |
| Major depressive disorder | 15 | 2 | 17 | |
| Dysthymic disorder | 1 | 2 | 3 | |
| Depression NOS | 8 | 7 | 15 | |
| Currently on antidepressants | | | | $\chi^2 = 0.50$ |
| Yes | 14 | 5 | 19 | |
| No | 10 | 6 | 16 | |
| Suicide attempts | | | | $\chi^2 = 7.91^{*}$ |
| Never | 21 | 5 | 26 | |
| Once | 2 | 2 | 4 | |
| More than once | 1 | 4 | 5 | |

Notes.
PS, psychache scale; OMMP, Orbach & Mikulincer current mental pain questionnaire; BDI, Beck depression inventory; BHS, Beck hopelessness scale; $D_x$, diagnosis; NOS, not otherwise specified.
*$p < .05$.
**$p < .01$.

**Table 3  Zero-order correlations between clinical symptoms by recent suicidal desire.**

| | 1 | 2 | 3 | 4 | 5 | 6 | 7[a] |
|---|---|---|---|---|---|---|---|
| 1. PS | – | $.97^{***}$ | $.80^{***}$ | $.50^{*}$ | $.78^{***}$ | $.50^{*}$ | $.21$ |
| 2. PS factor 1[b] | $.98^{***}$ | – | $.64^{***}$ | $.51^{*}$ | $.74^{***}$ | $.47^{*}$ | $.16$ |
| 3. PS factor 2[b] | $.82^{**}$ | $.70^{*}$ | – | $.35^{\dagger}$ | $.68^{***}$ | $.44^{*}$ | $.42^{*}$ |
| 4. OMMP | $.75^{**}$ | $.64^{*}$ | $.90^{***}$ | – | $.66^{***}$ | $.60^{**}$ | $.27$ |
| 5. BDI | $.57^{\dagger}$ | $.53^{\dagger}$ | $.57^{\dagger}$ | $.58^{\dagger}$ | – | $.54^{**}$ | $.58^{**}$ |
| 6. BHS | $.33$ | $.23$ | $.55^{\dagger}$ | $.52$ | $.64^{*}$ | – | $.51^{*}$ |
| 7. BSS[a] | $.07$ | $-.07$ | $.51$ | $.46$ | $.58^{\dagger}$ | $.57^{\dagger}$ | – |

Notes.
Group with recent suicidal desire ($n = 11$) shown below the diagonal and without recent suicidal desire ($n = 24$) above the diagonal.
PS, psychache scale; OMMP, Orbach & Mikulincer current mental pain questionnaire; BDI, Beck depression inventory; BHS, Beck hopelessness scale; BSS, Beck Scale for Suicide ideation minus item 4 to maintain independence.
[a]Spearman instead of Pearson correlations because of nonnormality.
[b]PS factor 1 contains nine frequency items, PS factor 2 contains four intensity items.
***$p < .001$.
**$p < .01$.
*$p < .05$.
†$p < .10$.

($r = .75$, $p < .01$ vs. $r = .50$, $p < .05$). Strikingly, the correlation between PS factor 2 and the OMMP score in the group with recent suicidal desire was very strong ($r = .90$, $p < .001$), but only a statistical trend existed for a correlation of medium strength in the group who reported no recent suicidal desire ($r = .35$, $p = .09$). The correlation between PS factor 1 and the OMMP score was strong for both groups (recent suicidal desire, yes: $r = .64$, $p < .05$; no: $r = .51$, $p < .05$). As the total PS score and factor scores are not

**Table 4  Simultaneous multiple regression models of psychological pain on right frontal EEG delta power controlling for level of depression and recent suicidal desire.**

|  | $F$ | $R^2$ | $b$ | $SE(b)$ | $\beta$ |
|---|---|---|---|---|---|
| PS | 19.13[***] | 68.1% |  |  |  |
|   BDI |  |  | 0.69 | 0.10 | .72[***] |
|   F4 delta |  |  | 5.59 | 2.18 | .28[*] |
|   SD |  |  | 4.38 | 2.32 | .20[†] |
|   SD × F4 delta |  |  | −14.04 | 4.98 | −.31[**] |
| PS factor 1[a] | 13.61[***] | 59.7% |  |  |  |
|   BDI |  |  | 0.06 | 0.01 | .68[***] |
|   F4 delta |  |  | 0.38 | 0.21 | .22[†] |
|   SD |  |  | 0.41 | 0.23 | .21[†] |
|   SD × F4 delta |  |  | −1.11 | 0.49 | −.28[*] |
| PS factor 2[a] | 13.39[***] | 59.3% |  |  |  |
|   BDI |  |  | 0.04 | 0.01 | .67[***] |
|   F4 delta |  |  | 0.53 | 0.17 | .38[**] |
|   SD |  |  | 0.18 | 0.18 | .12 |
|   SD × F4 delta |  |  | −1.01 | 0.39 | −.32[*] |
| OMMP | 8.25[***] | 46.0% |  |  |  |
|   BDI |  |  | 1.77 | 0.35 | .69[***] |
|   F4 delta |  |  | −11.61 | 7.57 | −.22 |
|   SD |  |  | 1.73 | 8.08 | .03 |
|   SD × F4 delta |  |  | −14.34 | 17.31 | −.12 |

**Notes.**

PS, psychache scale; BDI, Beck depression inventory; F4 delta, grand mean centered right midfrontal EEG delta power in $\ln(\mu V^2)$; SD, recent suicidal desire (yes/no); OMMP, Orbach & Mikulincer current mental pain questionnaire.

[a] PS factor 1 contains nine frequency items, PS factor 2 contains four intensity items.

[***] $p < .001$.

[**] $p < .01$.

[*] $p < .05$.

[†] $< .10$.

independent and testing of multiple dependent correlations inflates type I error (*Curtin & Schulz, 1998*), we also evaluated the data using a Bonferroni corrected significance level (.05/3 correlations). The correlations of the OMMP score with the total PS and factor 2 score in the group with recent suicidal desire remained significant. When we tested the association between PS factor 2 and OMMP in a linear model while controlling for the level of depression, the overall model was significant ($F = 9.08$, $df = 4, 30$, $p < .0005$) and a significant interaction of Group × PS factor 2 was found ($\Delta R^2 = .08$, $p < .05$). However, this interaction did not remain significant when using a Bonferroni correction for multiple tests. Similar models with the total PS score and PS factor 1 did not show a significant interaction ($\Delta R^2 = .02$, $p = .17$ for Group × PS; $\Delta R^2 = .00$, $p = .33$ for Group × PS factor 1). Effect sizes indicate that hypothesis 1 was supported primarily for factor 2 of the Psychache Scale. Small sample size precluded reaching significance levels in some of these analyses.

Findings with respect to hypothesis two are presented in Tables 4 and 5. Table 4 shows regression models of the PS score and PS factor scores on right frontal delta power, while controlling for depression and recent suicidal desire. A significant interaction of Group × delta power was observed ($\Delta R^2 = .07$, $p < .01$) with opposite associations between the PS score and delta power in the groups with ($\beta = -.37$, $p = .20$) and without

**Table 5** Simultaneous multiple regression models of psychological pain on low-frequency heart rate variability controlling for level of depression and recent suicidal desire.

| | $F$ | $R^2$ | $b$ | $SE(b)$ | $\beta$ |
|---|---|---|---|---|---|
| PS | 15.59*** | 63.2% | | | |
| BDI | | | 0.70 | 0.11 | .73*** |
| LF HRV | | | 2.57 | 1.28 | .24† |
| SD | | | 4.43 | 2.85 | .20 |
| SD × LF HRV | | | −5.24 | 3.40 | −.20 |
| PS factor 1[a] | 13.22*** | 59.0% | | | |
| BDI | | | 0.06 | 0.01 | .70*** |
| LF HRV | | | 0.26 | 0.12 | .28* |
| SD | | | 0.38 | 0.26 | .20 |
| SD × LF HRV | | | −0.44 | 0.31 | −.19 |
| PS factor 2[a] | 8.17*** | 45.8% | | | |
| BDI | | | 0.04 | 0.01 | .66*** |
| LF HRV | | | 0.06 | 0.11 | .08 |
| SD | | | 0.26 | 0.24 | .17 |
| SD × LF HRV | | | −0.32 | 0.29 | −.17 |
| OMMP | 10.59*** | 53.0% | | | |
| BDI | | | 1.50 | 0.33 | .59*** |
| LF HRV | | | −9.28 | 3.85 | −.32* |
| SD | | | 11.99 | 8.61 | .20 |
| SD × LF HRV | | | −11.36 | 10.26 | −.16 |

**Notes.**

PS, psychache scale; BDI, Beck depression inventory; LF HRV, grand mean centered low-frequency heart rate variability in $\ln(ms^2)$; SD, recent suicidal desire (yes/no); OMMP, Orbach & Mikulincer current mental pain.

[a] PS factor 1 contains nine frequency items, PS factor 2 contains four intensity items.

*** $p < .001$.

* $p < .05$.

† $p < .10$.

($\beta = .33$, $p < .01$) recent suicidal desire. Similar Group × delta power interaction effects were observed when the PS factor scores were regressed on delta power, with the interaction term accounting for $\Delta R^2 = .05$ ($p < .05$) in the model of PS factor 1 and $\Delta R^2 = .08$ ($p < .05$) in the model of PS factor 2. Table 4 also shows that no interaction effect was observed when the OMMP score was regressed on right frontal delta power.

Table 5 shows regression models of the PS score and PS factor scores on low-frequency HRV, while controlling for depression and recent suicidal desire. All overall models were significant, but none of the models showed a significant interaction of Group × low-frequency HRV. Post-hoc power analysis indicated a high risk for type II error ($\beta = .64$), given the size of the interaction effect for the model that included the PS total score (partial $\eta^2 = .073$). While statistically nonsignificant, all three models suggested opposite directions for the association between the psychological pain score and low-frequency HRV in the groups with and without recent suicidal desire (respectively, $\beta = -.22$, $p = .46$ and $\beta = .30$, $p < .05$ for the model that included the PS total score). No indication for an interaction effect was observed when the OMMP score was regressed on low-frequency HRV (see Table 5).

The difference in effect sizes for regression models of right frontal delta power and low-frequency HRV prompted us to explore the association between low-frequency HRV

and right frontal delta power. We regressed low-frequency HRV on right frontal delta power while controlling for depression, recent suicidal desire, and age, as age covaried with HRV. The overall model was significant ($F = 3.55$, $df = 3, 31$, $p < .05$), and right frontal delta power was positively associated with low-frequency HRV ($\beta = .36$, $p < .05$), confirming that the effect of recent suicidal desire on the association between low-frequency HRV and the PS score may have been nonsignificant due to a lack of power. There was no evidence of opposite associations between right-frontal delta power and low-frequency HRV for the groups with and without recent suicidal desire.

## DISCUSSION

The present study was a secondary analysis to test the moderating effect of recent suicidal desire on (1) correlations between the OMMP and the PS, and (2) on relationships between psychological pain and resting-state neurophysiological parameters. We found corroborating evidence for our hypothesis that the correlation between psychological pain as assessed on the OMMP and the PS depends on recent suicidal desire. This is especially clear for the pain intensity factor of the PS, which revealed a very strong positive association in participants with suicidal desire within the last week, while controlling for depression. In participants without recent suicidal desire the association between PS pain intensity (factor two) and OMMP was also positive, but of moderate strength and not statistically significant. For PS pain frequency (factor 1) and the total PS score, the differences were not as pronounced, but they too indicated considerably stronger associations for participants with recent suicidal desire.

Analysis of low-frequency HRV and right frontal EEG delta power provided evidence that recent desire for suicide moderates the relationship between the PS score and neurophysiological parameters. This was evident for EEG delta power, essentially confirming our second hypothesis, and to a lesser extent for low-frequency HRV. The significant positive correlation between low-frequency HRV and EEG delta power, while controlling for age and level of depressive symptoms, strengthens our interpretation that the difference in association between PS and low-frequency HRV in participants with and without recent suicidal desire would likely be statistically significant in a larger sample. Among participants with recent suicidal desire, EEG delta power and low-frequency HRV were both lower when individuals reported greater psychological pain on the PS. The direction and strength of these associations resemble findings in the complete sample for associations between OMMP and low-frequency HRV and EEG delta power, reported elsewhere (*Meerwijk, Chesla & Weiss, 2014*; *Meerwijk, Ford & Weiss, 2015*). In that study, decreased low-frequency HRV and EEG delta power were interpreted as indicators of less effective emotion regulation, including increased rumination and an inability to reappraise the causes and consequences of psychological pain. There is evidence that variability in resting-state heart rate and EEG reflects emotion regulation processes (*Beauchaine, 2001*; *Thayer et al., 2009*; *Bornas et al., 2012*; *Knyazev, 2012*). Rumination and cognitive reappraisal are recognized examples of emotion regulation strategies (*Aldao, Nolen-Hoeksema & Schweizer, 2010*). Interestingly, we found opposite results among
participants who reported no suicidal desire during the past week. In this group, low-frequency HRV and EEG delta power were higher when individuals reported greater psychological pain on the PS. We speculate that participants without recent suicidal desire may have had better emotion regulation skills. The majority of participants without recent desire for suicide (87.5%) had never attempted suicide, whereas a third of participants with recent suicidal desire had attempted suicide more than once. Research has shown that poor emotion regulation is a distinguishing factor between multiple attempters and individuals who attempted suicide once (*Reynolds & Eaton, 1986*; *Choi et al., 2013*). Moreover, rumination in psychiatric inpatients was positively associated with suicide ideation, and cognitive reappraisal was significantly higher in psychiatric inpatients who reported no suicide ideation (*Morrison & O'Connor, 2008*; *Forkmann et al., 2014*). However, the study by Forkmann et al. did not find differences based on suicidal desire.

The results corroborate our assumption that the PS and OMMP perform more similarly in the presence of recent or current psychological pain that is expected in individuals with recent suicidal desire. In contrast, the scales appear to diverge in individuals without recent suicidal desire. In this latter group, respondents may have reported on current pain in one measure (the OMMP) and on psychological pain over time or at a highly significant previous time in the other measure (PS). We postulate that a change in the instructions for the PS could improve concurrent validity of the two scales. The OMMP specifically instructs respondents to consider *current* psychological pain, whereas the PS leaves the time frame undefined. Although this may not be an issue in a population that is likely to experience psychological pain at the time of completing the PS, it does become relevant when study participants are not homogeneous with respect to when they experienced psychological pain, as was the case in our study. Given the robust relationship between suicide and psychological pain, it may be that participants who did not report a recent desire for suicide did not experience significant psychological pain recently either. We would expect the OMMP to capture this distinction, resulting in lower scores. Due to its more general instructions, these same participants may have completed the PS while reflecting on psychological pain they experienced at some undefined time in the past, resulting in higher scores than their current level of psychological pain warranted. It is likely that including a time frame in the measure's instructions, for example psychological pain during the past week, would address the issue. This would emphasize that the PS is about recent psychological pain and not about psychological pain experienced during some distant past. A similar instruction is used for the BSS, which assesses suicide ideation during the past week. If recency of psychological pain affects how individuals complete the PS, making the instrument's instructions more specific will enhance the PS's clinical applicability and facilitate comparison of PS scores between populations and comparison with correlates of psychological pain.

Alternatively, our results could reflect a genuine difference in sensitivity for suicidal desire in the items of the PS and OMMP, for which support can be found in the origin of the two scales. The PS items were derived from Shneidman's theory of suicide (*Holden et al., 2001*), whereas the OMMP items describe different aspects of psychological pain and were based on narratives and interviews with individuals familiar with psychological

pain due to aversive life events or personal issues (*Orbach et al., 2003a*). As such, it is not unreasonable to assume that the PS may be more sensitive to suicidal thoughts and behaviors. It is noteworthy that the PS score in the group with recent suicidal desire is significantly higher than in the group without recent suicidal desire, whereas the OMMP score did not differ significantly between the two groups. This could suggest that the PS and OMMP are sensitive to different underlying processes. Another obvious difference between the two scales is that the PS assesses both intensity and frequency of the pain, whereas the OMMP assesses intensity only. The impact of these different characteristics on assessment of psychological pain is not known.

Some limitations of this study should be addressed. The available sample size was small, especially for participants who reported a recent desire for suicide. Therefore, insufficient power may explain why some of our subgroup analyses produced nonsignificant results or reached trend-level significance only. However, despite a small sample size, we found significant interaction effects for suicidal desire, which suggests that the findings are unlikely to be a result of chance alone. Our sample was not homogeneous with respect to the participants' diagnosis of depression. To compensate for the differences in depression we controlled our analyses for the level of depressive symptoms. Participants knew they enrolled in a study about psychological pain. Although we did not disclose our specific hypotheses, participants' opinions about the purpose of the study may have affected how they completed the self-report instruments. However, our significant results involving neurophysiological parameters, which are objective by nature, suggest that findings are not likely the result of participant bias.

Our results suggest that recent suicidal desire moderates psychological pain assessed on the PS, whereas no such moderation existed for OMMP scores. We conclude that this moderating effect is not a genuine difference in the overall constructs being measured or sensitivity of the PS, but may be an artifact of the instrument's instructions with regard to the time frame when psychological pain was experienced. Our recommendation for changes to the instructions for completion of the PS could enhance its clinical applicability. Additional research is necessary to assess the effect of such a modification on concurrent validity of the PS and if such a modification leads to significant associations with resting-state neurophysiological parameters similar to those found previously for the OMMP.

### Funding

This study was supported by research grants to the first author from the American Psychiatric Nursing Foundation and Sigma Theta Tau International Honor Society of Nursing, Alpha Eta Chapter. The first author received additional support through the National Institute of Nursing Research Training Grant No. T32 NR07088. The co-author received additional support from the Robert C & Delphine Wentland Eschbach Endowment. The funders had no role in study design, data collection and analysis, decision to publish, or preparation of the manuscript.

## Grant Disclosures

The following grant information was disclosed by the authors:

American Psychiatric Nursing Foundation.

Sigma Theta Tau International Honor Society of Nursing, Alpha Eta Chapter.

National Institute of Nursing Research Training: T32 NR07088.

Robert C & Delphine Wentland Eschbach Endowment.

## Competing Interests

The authors declare there are no competing interests.

## Author Contributions

- Esther L. Meerwijk conceived and designed the experiment, performed the experiment, analyzed the data, wrote the paper, prepared figures and/or tables, reviewed drafts of the paper.
- Sandra J. Weiss conceived and designed the experiment, wrote the paper, reviewed drafts of the paper.

## Human Ethics

The following information was supplied relating to ethical approvals (i.e., approving body and any reference numbers):

Institutional Review Board at the University of California, San Francisco (IRB# 11-07119).

## Data Availability

The uploaded file Supplemental Information contains the data used for this analysis.

## Supplemental Information

Supplemental information for this article can be found online at http://dx.doi.org/10.7717/peerj.1538#supplemental-information.

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
