# Peer review of "Does suicidal desire moderate the association between frontal delta power and psychological pain?"

_PeerJ, doi:10.7717/peerj.1538_

## Round 0.1 · original submission · Minor Revisions

Please try to edit your text to include as many reviewers suggestions as possible and provide a full accounting of the changes made.

Reviewer 1 ·

Basic reporting

no comments

Experimental design

In general and as stated by the author, this is a secondary analysis of previously published study data. The hypotheses that 1) recent suicidal desire moderates the correlation between OMMP and PS scores and 2) the correlation between PS scores and resting state physiological parameters is moderated by recent suicidal desire are somewhat difficult as stated to follow. If the author is suggesting that recent suicidal desire is behind the correlations between two previously published psychological pain scales it appears to be another way of saying that psychological pain mediates the correlation between two psychological pain scales (that is if one accepts the commonly held and published conclusion that psychological pain is a powerful mediator of suicidality). The authors don’t provide adequate justification for the other scale measurements obtained except that depression and hopelessness correlate with psychological pain, which has of course been shown previously but there is little in the study design or hypothesis that mentions these other constructs. I would suggest language to justify their inclusion a priori such as ‘We included measures of depression and hopelessness to allow statistical analysis to remove the contribution of these constructs to correlations reported..” . I would like to see some kind of power analysis as well although given the admitted small sample size this may not be doable.

I cannot comment on the neurophysiological study design elements as I am not experienced in these procedures.

Validity of the findings

The 2nd hypothesis in a sense states that a correlation between one, but not the other, psychological pain scale and certain previously published neurphysiological parameters is mediated by psychological pain is interesting. That is, it is interesting to possibly have identified objective correlates of psychological pain, but as the authors report later in the paper, one would expect both scales to support this. The finding that the correlation between these two previously published scales was greater in recently suicidal groups may simply represent higher mean psychological pain intensity between these two groups---as we would expect.

In the discussion, the authors conclude that recent suicidality moderates the correlations between the OMMP and the PS, I’m not sure this relatively modest sample size and design is optimal for examining the underlying factors mediating correlation between two psychometric instruments. Again, if these instruments are acceptably validated and published then they would correlate on face value by nature of shared convergent validity.

The strongest data are the neurophysiological data and highlighting this correlation makes an interesting contribution, however I would like to see some discussion on hypothesized factors mediating this association-esp for readers not familiar with the parameters noted in the association. The results showing discordance in correlative results between the OMMP and PS are interesting and potentially important but certainly should be looked at in other studies. The discussion as presented on what may underlie this discordance is reasonable although there may be, of course, a number of other factors that are at play. It would add to the discussion to put forth briefly a few other possible explanations.

Additional comments

Overall, I think simply restructuring or rewording the hypothesis, emphasizing some points in the data over others and avoiding over-reach with this modest sample size would improve the paper but I believe this is publishable with some relatively minor re-working. Again, the neurophysiological data are potentially the strongest points and probably best made an emphasis of the paper.

Reviewer 2 ·

Basic reporting

The authors reported original research findings from their investigation testing the hypothesis if recent suicidal desire can moderate the association between frontal delta power and psychological pain, pointing to the role of psychache scale. The paper is of interest for the journal. Whi

Experimental design

While the experimental design is adequate, authors should provide more information on how patients were enrolled, the time frame of this and the procedure to administering tests, such who approached the patients, etc

Validity of the findings

Findings are important for contributing to the body of literature on the topic. I would encourage to expplore even further the matter and developing a bit more the discussion by translating their findings into the real of suicide attempts as related to life events which are a source of psychological pain; by citing a paper such as Pompili et al. Life events as precipitants of suicide attempts among first-time suicide attempters, repeaters, and non-attempters. Psychiatry Res. 2011

Additional comments

This is a nicely written paper that may be improved following comments

Reviewer 3 ·

Basic reporting

This paper essentially is studying the relationship between two instruments that measure psychological pain. The aim is to try to account for differences in them by looking at "recent suicidal desire" and some psychophysiological measures. It is hypothesized that psychophys measures would also account for variance. The introduction lacks any background as to why there would be a relationship between the psychophysiological measures and psychological pain.

Experimental design

The authors believe that recent suicidality as measured by the SD scale, is likely to strengthen the relationship between the Orbach (OMMP) and Psychache Scale (PS) because the PS has no timeframe associated with it but recent suicidality would likely be captured. This question would be better addressed using a more objective measure of suicidality. The Columbia SS can look at lifetime suicidality as well as recent (last 4 weeks) suicidality. This could represent a superior approach to looking at current versus lifetime issues. As is reported, ''suicidal desire" is based on a single item on the BSS.

There is no discussion of the diagnostic assessments used to determine diagnosis. The subject pool is peculiar, generally a single unemployed group of depressed individuals. It is important to know what the diagnosis is at the time of enrollment.

Validity of the findings

Many correlations are made but there is no explicit discussion of making adjusting p values for multiple comparison. This is particularly concerning given the small sample size. The authors are referred to the following reference regarding corrections for multiple correlations (Curtin and Schulz, 1998).

The authors conclude that suicide desire accounts for some differences between the scales. They then state that the OMMP is better suited for capturing current psychological pain more effectively than the PS. This seems to simply confirm what is outlined in the introduction ... that "OMMP specifically assesses current psych pain ... " versus the PS that "leaves timeframe undefined".

The authors don't consider other candidates such as other kinds of current stress that could mediate differences between the scales. For example what if a person had experienced a loss? There are many things likely to increase current psychological pain that are not necessarily directly related to suicidality.

The BSS correlations with the PS2 (intensity) and OMMP are nearly identical. The same can be said for the BDI. This does support the conclusions that frequency of psych pain (PS1) may not be as useful.

---

## Round 0.2 · accepted · Accept

All the points raised by the reviewers have been sufficiently addressed.